# Scalable Traffic Signal Control with Shared Policy Framework

**Haolun Ma** [1]   **Yanchen Zhu** [1]   **Zizhuo Xu** [2]   **Weijie Shi** [2]   **Jiajie Xu** [3]   **Lei Li** [1 2]

## Abstract

Learning-based Traffic Signal Control (TSC) achieves satisfactory performance in small networks, but its effectiveness often deteriorates in larger networks under dynamic traffic patterns and intersection heterogeneity. In this work, we propose SLight, a policy-aware grouped MARL-TSC framework that enables scalability and efficiency balance under dynamic and heterogeneous traffic conditions. SLight captures policy-influenced traffic patterns with a policy-aware traffic pattern encoder, learns explicit group-level shared control principles from state–action trajectories, and matches each intersection's traffic pattern embedding to principle prototypes flexibly through a compatibility-based adaptive assignment module. Experiments on real-world and synthetic networks demonstrate that SLight sustains performance gains as scale increases and outperforms existing rule-based, reinforcement learning, and grouping-based baselines. Code is available at https://github.com/MaHaoLun/Slight-code.git

## 1. Introduction

With the rapid growth of urban traffic, modern cities are suffering from more severe congestion (Papageorgiou et al., 2003; Systematics et al., 2005) caused by the increasing volume of traffic flows, especially around the intersections where different roads' traffic converge, diverge, meet, or cross. To determine how vehicles move and queue at these intersections, Traffic Signal Control (TSC) (Gartner, 2001) is introduced to regulate the phase switching and green time durations of traffic lights, with the goal of minimizing

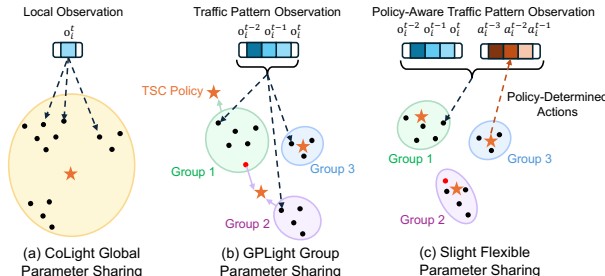

*Figure 1.* Comparison of Co-Light, GPLight, and our SLight

vehicle travel time and waiting time (Webster, 1958) to improve the overall traffic throughput.

In real-world traffic systems, TSC has long been dominated by rule-based and optimization-based methods. Representative approaches such as FixedTime (Koonce et al., 2008) and Maxpressure (Varaiya, 2013) rely on predefined signal plans or local pressure-based rules to regulate phase switching and green time allocation. Although they are simple, interpretable, and easy to deploy, their fixed and local control strategies limit their ability to cope with the complex and highly dynamic traffic patterns.

To handle the complex and dynamic traffic flexibly, Reinforcement Learning (RL) has been introduced to TSC, enabling agents to learn adaptive policies based on the current traffic conditions. Approaches like FRAP (Zheng et al., 2019) and PressLight (Wei et al., 2019a) achieve satisfactory performance in small-scale networks, but their effectiveness often deteriorates when deployed to large networks due to non-stationary traffic dynamics, coordination complexity, and training instability. Recently, Large Language Model (LLM) has been introduced (LLMlight (Lai et al., 2024), Traffic-R1 (Zou et al., 2025)) to TSC by leveraging its semantic reasoning and zero-shot capabilities. However, they have not been validated and can hardly scale to large networks due to their slow training and high computational costs, which further limit their practical deployment.

To scale on large networks, Multi-Agent Reinforcement Learning (MARL) has been introduced to TSC by coordinating multiple traffic signals to make decisions together. However, as the agent number grows, the dimensions of the joint action/state space grow exponentially, which makes the

[1]Data Science and Analytics Thrust, Hong Kong University of Science and Technology (Guangzhou), Guangzhou, China [2]Department of Computer Science and Engineering, Hong Kong University of Science and Technology, Hong Kong SAR, China [3]School of Computer Science and Technology, Soochow University, Suzhou, China. Correspondence to: Lei Li <thorli@ust.hk>.

*Proceedings of the 43rd International Conference on Machine Learning*, Seoul, South Korea. PMLR 306, 2026. Copyright 2026 by the author(s).

training hard to converge. To reduce the number of learnable parameters, CoLight (Chen et al., 2020) and MPLight (Chen et al., 2020) adopt global parameter sharing across all agents (Fig. 1-(a)), but they overlook the behavioral heterogeneity among intersections with different traffic roles, so their performance suffers. To balance scalability and effectiveness of MARL-based TSC at the city scale, GPLight (Liu et al., 2023) introduces agent grouping, where agents with similar characteristics are clustered and share a common policy, as illustrated in Fig. 1-(b). But it still faces the following fundamental limitations:

1) *Insufficient Observations*. As illustrated in Fig.1-(a), CoLight only captures the current traffic conditions at each intersection and its neighboring ones. To capture the traffic patterns for better coordination, GPLight expands the horizon to further observe the near-history traffic conditions (Fig.1-(b)). However, because the traffic condition is also influenced by the trained signal policy, only relying on the observed traffic condition is not sufficient and could lead to the mismatch between embeddings and policies: intersections that look similar in the embedding space can still exhibit different temporal behaviors during learning, limiting the expressiveness of shared policies.

2) *Irrelevant Grouping*. Parameter sharing is effective only when the grouped intersections align with the trained policies. However, GPLight first clusters the intersections into groups only based on the embedding similarity, and then assigns these groups to different policies, which are irrelevant to each other. As a result, as illustrated in Fig.1-(b), some intersections are assigned to the wrong policy (red intersection in Group 1), and some groups are still far away from their most suitable policy.

3) *Passive Assignment*. When the traffic condition changes, the observation together with the grouping also changes, so the intersections should be assigned to new groups with corresponding new suitable policies. Although GPLight supports such dynamicity, its reassignment is still restricted by the feature-space proximity (dashed arrows) rather than whether the currently shared policy remains suitable as policies and traffic regimes evolve. This can mis-route an intersection to a visually similar but behaviorally mismatched group. What's worse, it also prolongs the overall training time and deteriorates effectiveness.

The root causes of the above limitations are the missing connection between the intersection embedding and policy, and the tight coupling of the intersection observation with the control policy. Therefore, to break these limitations and enable a more effective MARL-based TSC, we propose a novel framework **SLight** with three corresponding components: (1) A *Policy-Aware Traffic Pattern Encoder* that establishes the relation between traffic patterns and policies by encoding each intersection's spatial-temporal evolution

under the current policy (new red observations in Fig.1-(c)). It should be noted that, unlike the existing MARL-TSC, these embedded intersections will not be clustered to form unstable and irrelevant policy groups, but will be matched to the most suitable policy identified by the (2) *Shared Control Principle Representation* that decouples the grouping of the control policies from the intersections to form a more stable and general grouping (stars in Fig.1-(c)). It makes the overall framework faster to converge and achieve better performance. Finally, the (3) *Adaptive Assignment Module* maps the embedded intersections dynamically to their most suitable control policy based on the current traffic conditions by treating the group selection as a policy-aware decision process rather than a passive clustering output. In this way, the semantic drift caused by similarity-only reassignment is mitigated. Our contributions are summarized below:

- We propose **SLight**, a policy-aware grouped MARL-TSC framework that enables scalability and efficiency balance under dynamic and heterogeneous traffic conditions.

- SLight improves inner-group behavioral coherence and training stability by (i) capturing the policy influence on traffic with a *policy-aware traffic pattern encoder*, (ii) learning the shared control policies separately from intersections, and (iii) performing policy-aware matching to update group membership under *adaptive assignment*.

- We conduct extensive experiments on real and synthetic networks to evaluate the effectiveness, scalability, and interpretability of SLight, revealing improvements over existing methods.

## 2. Related Work

Existing TSC methods can be categorized into three types. **1) Static TSC.** FixTime (Koonce et al., 2008) relies on fixed phase cycles and phase allocations set by human experts, while MaxPressure (Varaiya, 2013) prioritizes phases according to traffic pressure. Both methods are simple and efficient, but they are hard to adapt to dynamic and heterogeneous traffic conditions. **2) RL-based TSC.** FRAP (Zheng et al., 2019) proposed phase competition to capture phase relations and better handle unbalanced traffic; CoLight (Wei et al., 2019b) leveraged graph attention to enable cooperative decision-making; PressLight (Wei et al., 2019a) incorporated pressure into the reward signal, showing that physically grounded congestion indicators can substantially improve RL performance; MPLight (Chen et al., 2020) extended FRAP to metropolitan-scale control by combining phase competition with pressure-aware signals; Efficient-CoLight (Wu et al., 2021) further improved state abstraction using efficient-pressure representation; and Advanced-CoLight (Zhang et al., 2022) demonstrated that compact expressions of pressure and demand can serve as effective

state representations for GNN-based RL controllers. Recent methods further improve adaptability, coordination, and deployability from different perspectives. MetaVIM (Zhu et al., 2023) introduces meta-RL for decentralized TSC under changing environments; TransformerLight (Wu et al., 2023) explores Transformer-based sequence modeling for traffic signal control; CoSLight (Ruan et al., 2024) co-optimizes collaborator selection and decision-making; GPLight+ (Liao et al., 2025) uses genetic programming to learn symmetric signal-control policies; and Proactive-XLight (Jiang et al., 2025) incorporates traffic prediction for proactive traffic signal control. **3) LLM/VLM-based TSC.** LLM-based and VLM-based methods, such as LLM-Light (Lai et al., 2024), Traffic-R1 (Zou et al., 2025), and VLMLight (Wang et al., 2025), explore semantic reasoning, instruction following, and vision-language meta-control for traffic signal control. These methods provide promising generalization mechanisms, while their training and inference costs remain important considerations for large-network deployment.

Traffic simulators are essential for developing and evaluating TSC methods because real-world deployment is costly, risky, and hard to reproduce. SUMO (sum, 2023) provides detailed microscopic dynamics and flexible control, whereas CityFlow (Zhang et al., 2019) is optimized for large-scale RL with high simulation throughput. Commercial simulators such as VISSIM (Fellendorf & Vortisch, 2010) and Aimsun (Barceló & Casas, 2005) offer high fidelity but are often less suitable for large-scale learning due to closed-source constraints, while MATSim (Axhausen et al., 2016) supports city-scale mesoscopic/macroscopic modeling with reduced control granularity. Thus, recent learning-based TSC studies commonly adopt open, programmable simulators such as SUMO and CityFlow to balance realism, scalability, and experimental reproducibility.

# 3. Preliminaries

## 3.1. Road Network and Signal Control Elements

**Road Network.** We model a road network as a directed graph $G = (V, E)$, where each node $i \in V$ is a signalized intersection and each directed edge $(u, v) \in E$ is a road segment. For intersection $i$, let $\mathcal{L}_i^{\text{in}}$ and $\mathcal{L}_i^{\text{out}}$ denote its incoming and outgoing lanes. Lane-level traffic variables are measured on $\mathcal{L}_i^{\text{in}}$.

**Movements and Phases.** Vehicles traverse an intersection via *movements* (left/through/right) from an incoming road to an outgoing road. Because there are twelve different incoming roads with different directions, we have twelve movements. For a vehicle in one specific incoming lane, its outgoing road is determined, but the detailed lane is not. For example in Fig.2-(a), a vehicle in lane B of Intersection

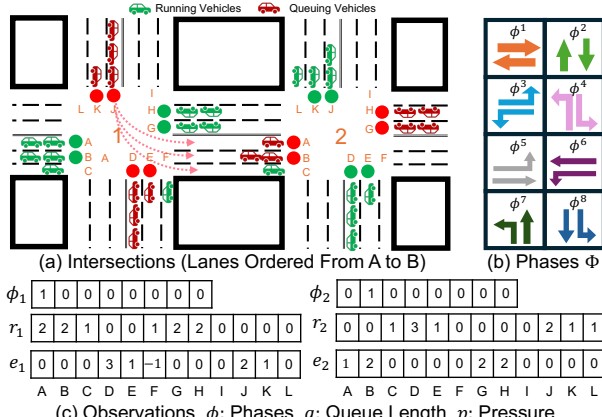

(a) Intersections (Lanes Ordered From A to B)    (b) Phases Φ

| $\phi_1$ | 1 | 0 | 0 | 0 | 0 | 0 | 0 | 0 | | $\phi_2$ | 0 | 1 | 0 | 0 | 0 | 0 | 0 | 0 |
|---|---|---|---|---|---|---|---|---|---|---|---|---|---|---|---|---|---|---|
| $r_1$ | 2 | 2 | 1 | 0 | 0 | 1 | 2 | 2 | 0 | 0 | 0 | 0 |
| $e_1$ | 0 | 0 | 0 | 3 | 1 | −1 | 0 | 0 | 0 | 2 | 1 | 0 |

| $r_2$ | 0 | 0 | 1 | 3 | 1 | 0 | 0 | 0 | 0 | 2 | 1 | 1 |
| $e_2$ | 1 | 2 | 0 | 0 | 0 | 0 | 2 | 2 | 0 | 0 | 0 | 0 |

A B C D E F G H I J K L     A B C D E F G H I J K L

(c) Observations. $\phi$: Phases. $q$: Queue Length. $p$: Pressure

*Figure 2.* Example Intersections with Observations

1 can only enter road $(1, 2)$ because lane B is a through lane, but we do not know which lane (A, B, or C of Intersection 2) it will enter. A *signal phase* is a set of non-conflicting movements that can be simultaneously granted right-of-way. In this work, we assume the right-turn is always possible, so there are eight phases in total (Fig.2-(b)), denoted by $\Phi_i = \{\phi_i^1, \ldots, \phi_i^8\}$.

**Control Interval.** Time is discretized into steps of length $\Delta t$. At each step $t$, the controller selects the next phase, which remains active during $[t, t + \Delta t)$. Following standard TSC benchmarks, we focus on *phase selection* with a fixed $\Delta t$ to isolate scalability and coordination in large networks.

## 3.2. Multi-Agent Traffic Signal Control Formulation

**Agents and Joint Interaction.** With the above signal elements, we model each intersection $i \in V$ as an agent, and denote the number of agents by $N = |V|$. Agents interact through coupled traffic dynamics, since upstream decisions affect downstream queues and flows.

**Observations.** At time $t$, intersection $i$ receives a local observation $o_i^t$ summarizing its current signal status and incoming-lane traffic. Concretely, we use (i) a one-hot phase indicator $\phi_i^t \in \{0, 1\}^{|\mathcal{P}_i|}$ (as illustrate in the first row of Fig.2-(c)), and (ii) two lane-level traffic features: the *effective running vehicle number* $r_i^t(l)$ and the *efficient pressure* $e_i^t(l, m)$, which are defined below:

**Definition 3.1** (**Effective Running Vehicle Number** $r_i^t(l)$). For each incoming lane $l \in \mathcal{L}_i^{\text{in}}$, the effective running vehicle number is the number of running vehicles (green vehicles in Fig.2-(a)) within the effective range to the intersection:

$$r_i^t(l) = \sum_{x \in \mathcal{X}_i^t(l)} \mathbb{I}\left[v_x^t > 0\right], \quad (1)$$

where $\mathcal{X}_i^t(l)$ is the set of vehicles located in the effective range of lane $l$ at time $t$, $v_x^t$ is the speed of vehicle $x$, and

$\mathbb{I}[\cdot]$ is the indicator function. Fig.2-(c)'s second row shows $r_i^t(l)$ of all lanes.

**Definition 3.2** (**Efficient Pressure** $e_i^t(l, m)$)**.** For each traffic movement $(l \to m)$ from an incoming lane $l$ to a downstream (outgoing) lane/group $m$, the efficient pressure is the average queue-length difference between upstream and downstream:

$$e_i^t(l, m) \;=\; \frac{1}{M(l)} \sum_{l' \in \mathcal{L}(l)} q_i^t(l') \;-\; \frac{1}{N(m)} \sum_{m' \in \mathcal{L}(m)} q_i^t(m'), \quad (2)$$

where $q_i^t(\cdot)$ is the queue length (red queuing vehicles number) on a lane, $\mathcal{L}(l)$ and $\mathcal{L}(m)$ are the sets of used lanes associated with road $l$ and $m$, and $M(l) = |\mathcal{L}(l)|$, $N(m) = |\mathcal{L}(m)|$. For example, $e_1^J = 2$ because its queuing vehicles number 3 minus the downstream road's average queuing vehicle number of 3/3=1 is 2. $e_1^F = -1$ because it has no queuing vehicle.

Finally, we form the local observation by concatenation:

$$o_i^t = \left[ \phi_i^t; \; \{r_i^t(l)\}_{l \in \mathcal{L}_i^{\text{in}}}; \; \{e_i^t(l, m)\}_{(l \to m) \in \mathcal{M}_i} \right], \quad (3)$$

where $\mathcal{M}_i$ denotes the set of feasible traffic movements at intersection $i$. The joint observation of all intersections is $\mathbf{o}^t = (o_1^t, \ldots, o_N^t)$.

**Actions.** Agent $i$ selects an action $a_i^t \in \Phi_i$, where $\mathcal{A}_i = \{1, \ldots, |\Phi_i|\}$ indexes the available phases and $a_i^t = k$ activates phase $\phi_i^k$ for the next interval. The joint action of all agents is $\mathbf{a}^t = (a_1^t, \ldots, a_N^t)$.

**Traffic Pattern.** Traffic pattern of intersection $i$ at time $t$ is a window of recent observations:

$$\mathbf{s}_i^t = \{o_i^{t-\tau+1}, \, o_i^{t-\tau+2}, \, \ldots, \, o_i^t\}, \quad (4)$$

where $\tau$ is the window length.

**Dynamics and Cooperative Objective.** Traffic evolution induces a (partially observable) transition $\mathbf{o}^{t+1} \sim \mathcal{P}(\cdot \mid \mathbf{o}^t, \mathbf{a}^t)$. Each agent receives an immediate reward $r_i^t$ computed from local traffic conditions (*e.g.,* negative queue length or delay). The goal of TSC is cooperative: learn policies that maximize the expected discounted average return over all intersections,

$$\max \; \mathbb{E}\left[ \sum_{t=0}^{\infty} \gamma^t \cdot \frac{1}{N} \sum_{i=1}^{N} r_i^t \right], \quad (5)$$

where $\gamma \in [0, 1]$ is the discount factor.

### 3.3. Shared-Policy Setting and Traffic Patterns

**Policies.** Given the observation and action definitions in Sec. 3.2, a policy specifies how an intersection selects a signal phase based on its observation. Concretely, for each

intersection $i$, a policy $\pi_\theta$ parameterized by $\theta$ maps $o_i^t$ to a distribution over the phase indices $\mathcal{A}_i = \{1, \ldots, |\Phi_i|\}$:

$$a_i^t \sim \pi_\theta(\cdot \mid o_i^t), \qquad a_i^t \in \mathcal{A}_i, \quad (6)$$

where $a_i^t = k$ activates phase $\phi_i^k \in \Phi_i$ for the next interval. Under this formulation, learning TSC amounts to optimizing policy parameters so that the induced joint actions $\mathbf{a}^t$ improve the cooperative objective in Eq. (5).

**Grouped Policies and Assignment.** Let $\{\pi_{\theta_1}, \ldots, \pi_{\theta_K}\}$ denote $K$ group policies. At each time step, intersection $i$ is assigned a group label $z_i^t \in \{1, \ldots, K\}$ and acts according to the corresponding policy:

$$a_i^t \sim \pi_{\theta_{z_i^t}}(\cdot \mid o_i^t). \quad (7)$$

This formulation decouples *policy capacity* from *network scale*: the policy set size $K$ controls the model/compute budget, while the assignment variables $\{z_i^t\}$ determine how heterogeneous intersections are covered by the shared policies. If $z_i^t$ is fixed, the formulation reduces to *fixed grouping*; if $z_i^t$ is allowed to change, it becomes *adaptive assignment*, which is the setting studied in this paper. Importantly, effective sharing depends not only on defining $K$, but also on ensuring that the assignment mechanism matches intersections to policies that induce compatible control behaviors.

## 4. SLight Framework

In this section, we introduce our *SLight*, whose pipeline is illustrated in Fig.3. Different from the existing MARL-TSC, we decouple the grouping of TSC policy from the grouping of the intersection's traffic pattern. This is based on the fact that no matter how the traffic pattern changes, the set of $K$ underlying signal policies that we can choose from should be the same and learned. Therefore, from now on, the grouped control policies are called *principles*, and their representations are learned in the upper orange *shared control principle representation* component. The traffic patterns, together with policy influence, of each intersection are encoded by the lower blue *policy-aware traffic pattern encoder*, which we use as a *traffic query* for subsequent assignment rather than a standalone group identity. Then each intersection is mapped to one of the control principles with green *adaptive assignment*. The final policy network is applied to the simulation environment, which returns observations and rewards for traffic pattern encoding and control principle training. In the following, we will elaborate on the details of these components.

### 4.1. Policy-aware Traffic Pattern Encoder

In large-scale traffic signal control, a traffic-pattern representation should reflect not only how an intersection evolves over time, but also how this evolution is *influenced by recent control actions*. To this end, we design a policy-aware

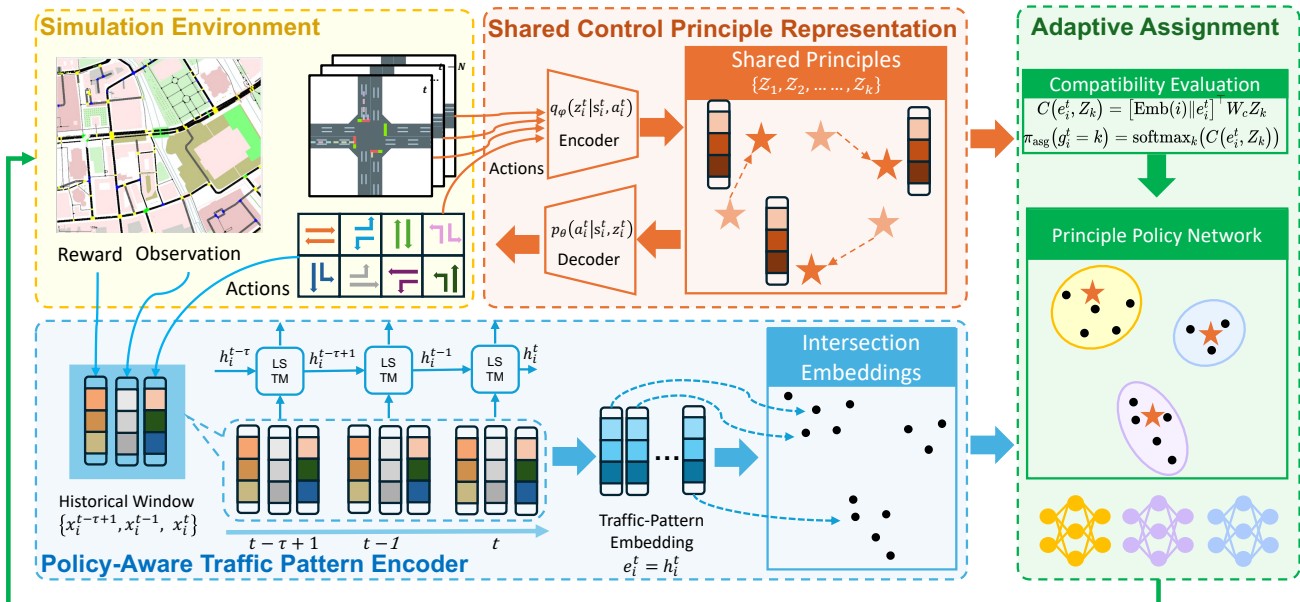

*Figure 3.* **SLight framework.** Policy-aware traffic pattern encoder: extract a traffic-query embedding $e_i^t$ per intersection; Shared control principle representation: learn $K$ principle prototypes $\{Z_k\}$ as grouped policies; Adaptive assignment: match $e_i^t$ to $\{Z_k\}$ for routing and control; Simulator loop: execute actions and return observations/rewards for joint updates.

temporal encoder that summarizes a short history of policy-aware transitions into a compact agent embedding for downstream grouping and assignment.

**4.1.1 Policy-Aware Traffic Patterns.** In Sec. 3.2, the traffic pattern of intersection $i$ at time $t$ is defined as an observation window $\mathbf{s}_i^t$ in Eq. (4). To make the pattern representation aligned with the *current policy*, this context is extended into a policy-aware interaction window by associating each observation with the most recent executed phase decision:

$$\tilde{\mathbf{s}}_i^t = \{(o_i^{t-\tau+1}, a_i^{t-\tau}), \ldots, (o_i^t, a_i^{t-1})\}, \qquad (8)$$

where $\tau$ is the window length. This policy-aware context provides short-horizon evidence of how local traffic evolves *under recent control actions*. The temporal encoder instantiates Eq. (8) online by converting each interaction step into a token $x_i^t$ (Eq. (9)) and aggregating the resulting sequence into a single embedding.

**4.1.2 Policy-Aware Temporal Encoding.** Given the policy-aware interaction window $\tilde{\mathbf{s}}_i^t$ in Eq. (8), the encoder operates online by mapping each step to a per-step policy-aware input. At time step $t$, intersection $i$ observes $o_i^t$; the most recent executed control action is $a_i^{t-1}$, and its immediate outcome is summarized by the one-step reward $r_i^{t-1}$ defined in Sec. 3.2. The per-step input is constructed as

$$x_i^t = \left[ o_i^t \,\|\, \rho(a_i^{t-1}) \,\|\, r_i^{t-1} \right], \qquad (9)$$

where $\rho(\cdot)$ encodes the discrete phase action. Including $r_i^{t-1}$ is necessary because action alone cannot evaluate their

*realized* control effects: the same phase decision can yield different observation changes under different local traffic conditions. The reward term provides a compact feedback channel aligned with the optimization objective (Eq. (5)), allowing the temporal representation to distinguish not only *what* was executed but also *how it affected* local traffic.

Then, these policy-aware inputs are aggregated into a temporally consistent summary by an LSTM over the window $\{x_i^{t-\tau+1}, \ldots, x_i^t\}$:

$$(h_i^t, c_i^t) = \text{LSTM}(x_i^t, h_i^{t-1}, c_i^{t-1}), \qquad (10)$$

whose gated memory retains persistent effects induced by recent control actions while suppressing transient fluctuations. As a result, $h_i^t$ encodes short-horizon traffic evolution under the current controller rather than merely reflecting snapshot similarity.

**4.1.3 Policy-Aware Traffic-Pattern Embedding.** We use the LSTM hidden state as the traffic-pattern embedding, *i.e.*, $e_i^t = h_i^t$. This is because $h_i^t$ aggregates the per-step policy-aware inputs $\{x_i^{t-\tau+1}, \ldots, x_i^t\}$ that instantiate $\tilde{\mathbf{s}}_i^t$, then $e_i^t$ captures the *policy-conditioned temporal evolution* of local traffic rather than snapshot similarity. However, this embedding $e_i^t$ is very unstable during learning, because policy updates change the induced action history and traffic feedback in $\tilde{\mathbf{s}}_i^t$, shifting the embedding distribution and potentially causing frequent reassignment if grouping were based on $e_i^t$ alone. This instability can make direct grouping on traffic embeddings sensitive to policy updates, especially

in large networks with coupled dynamics. To solve this problem, SLight uses $e_i^t$ only as a *traffic query* for adaptive assignment, while introducing *Shared Control Principle Representation*.

## 4.2. Shared Control Principle Representation

In Sec. 4.1, SLight treats $e_i^t$ only as a *traffic query* for assignment, and additionally learns an explicit group-level representation—*shared control principles*—that summarizes *how a shared policy reacts* to traffic states. These principles act as slowly-varying behavioral references, enabling stable grouping and effective parameter sharing even when traffic queries drift.

### 4.2.1 From Policy-Aware Traffic Patterns to Control Principles.
Recall that the policy-aware traffic pattern is defined as the interaction window $\tilde{\mathbf{s}}_i^t = \{(o_i^{t-\tau+1}, a_i^{t-\tau}), \ldots, (o_i^t, a_i^{t-1})\}$, whose encoding yields $e_i^t$; notably, $\tilde{\mathbf{s}}_i^t$ itself is a sequence of policy-induced state–action interactions (with $o_i^t$ or its encoded features $s_i^t$, and action $a_i^t$). While $e_i^t$ emphasizes *how traffic evolves*, effective parameter sharing further requires *consistent decision tendencies*—i.e., under similar policy-aware traffic patterns, the shared policy should respond with similar action selections. We therefore introduce a latent variable $z_i^t$ to distill such decision tendency from these interactions, and refer to it as a *control principle* that captures the shared policy's response pattern beyond state similarity.

### 4.2.2 Learning Structured Latent Control Principles via CVAE.
To learn *control principles* that summarize *group-level decision tendencies* (rather than traffic states), we train a Conditional Variational Autoencoder (CVAE) on the policy-aware interactions within $\tilde{\mathbf{s}}_i^t$. For each step $t$ in the window, we use the traffic-pattern state $\mathbf{s}_i^t$ and the executed discrete action $a_i^t$ as input: the *principle encoder* infers a posterior over the latent principle, and the *principle-conditioned decoder* reconstructs the action distribution,

$$\mathcal{L}_{\text{CVAE}}(i, t) = \mathbb{E}_{q_\phi(z_i^t | \mathbf{s}_i^t, a_i^t)}\big[\log p_\theta(a_i^t \mid \mathbf{s}_i^t, z_i^t)\big]$$

$$-\text{KL}\big(q_\phi(z_i^t \mid \mathbf{s}_i^t, a_i^t)\,\|\,p(z)\big),$$

where we use a categorical likelihood for discrete phase actions. Conditioning on $\mathbf{s}_i^t$ encourages $z_i^t$ to capture *action-selection patterns* (policy responses) beyond state similarity, while the KL term regularizes the latent space into compact and comparable regions, making it suitable for forming *group representations*. Compared with clustering traffic queries $e_i^t$ directly, CVAE-learned principles are *behavior-grounded* in the decisions within $\tilde{\mathbf{s}}_i^t$, which reduces within-group action divergence under sharing and provides a more stable basis for adaptive assignment.

### 4.2.3 Forming Shared Control Principle Representations.
We maintain $K$ shared control principles $\{Z_k\}_{k=1}^K$, each corresponding to one shared policy (group). During training, each intersection produces a traffic query $e_i^t$ (Sec. 4.1) and is routed to a group $g_i^t$ by adaptive assignment; meanwhile, we store the resulting policy-aware traffic patterns into the replay buffer together with their group index $g_i^t$. Periodically, we replay buffered samples and map them to latent codes through the CVAE encoder, using the posterior mean $\mu_\phi(\mathbf{s}_i^t, a_i^t)$ as the principle embedding. For each group $k$, we update its principle representation by aggregating the latent means of samples with $g_i^t = k$ (*e.g.,* momentum averaging):

$$Z_k \leftarrow (1 - \alpha)Z_k + \alpha \cdot \text{Avg}\big(\{\mu_\phi(\mathbf{s}_i^t, a_i^t) \mid g_i^t = k\}\big),$$

which yields a stable group-level summary of the recurring decision tendency implemented by the $k$-th shared policy. Since $\{Z_k\}$ are updated from replayed trajectories with smoothing, the principle space evolves *gradually* with learning and remains identifiable, thereby stabilizing group membership updates under drifting traffic queries.

## 4.3. Adaptive Assignment Module

Given the shared control principles $\{Z_k\}_{k=1}^K$, we assign each intersection $i$ to one of the $K$ shared policies based on its current traffic query $e_i^t$ while keeping assignments stable during learning. Specifically, we compute a compatibility score between $e_i^t$ and each principle $Z_k$ via a lightweight bilinear form with an identity-aware projection:

$$s_{ik}^t = C(e_i^t, Z_k) = \left(W_g[\text{id}(i)\|e_i^t]\right)^\top Z_k,$$

$$\pi_{\text{asg}}(g_i^t{=}k) = \text{softmax}_k(s_{ik}^t).$$

where $\text{id}(i)$ is a one-hot identity feature used only in the assignment module to stabilize routing. It does not create intersection-specific policies: all intersections are still routed to one of the same $K$ shared Q-networks, and the principles $\{Z_k\}$ are learned from replayed trajectories rather than from identifiers. Thus, the assignment is anchored by persistent intersection context but remains driven by the compatibility between the traffic query $e_i^t$ and the learned principle $Z_k$.

We select $g_i^t$ by $\epsilon$-greedy sampling from $\pi_{\text{asg}}$ during training and by $\arg\max_k \pi_{\text{asg}}(g_i^t{=}k)$ during evaluation. The selected group determines the shared Q-network that produces the low-level action $a_i^t$ (*e.g.,* phase selection). The resulting policy-aware samples are stored with their group indices and replayed to update the matched principle $Z_{g_i^t}$, maintaining alignment between group assignment and shared-policy behavior as traffic queries drift over time.

| Method | JN | | | | | | | | | HZ | | | | | | NY1 | | |
|---|---|---|---|---|---|---|---|---|---|---|---|---|---|---|---|---|---|---|
| | 1 | | | 2 | | | 3 | | | 1 | | | 2 | | | 1 | | |
| | ATT | AWT | TP | ATT | AWT | TP | ATT | AWT | TP | ATT | AWT | TP | ATT | AWT | TP | ATT | AWT | TP |
| *Rule-based methods* | | | | | | | | | | | | | | | | | | |
| **FixedTime** | 443.17 | 429.22 | 5766 | 382.71 | 382.42 | 4285 | 398.27 | 398.52 | 5144 | 505.73 | 502.87 | 2658 | 414.78 | 412.69 | 4369 | 1514.21 | 879.56 | 6824 |
| **MaxPressure** | 288.36 | 288.85 | 6138 | 248.98 | 250.29 | 4319 | 248.72 | 251.38 | 5371 | 287.51 | 290.43 | 2927 | 327.73 | 322.38 | 5422 | 1129.26 | 918.22 | 7844 |
| *Unstructured RL-based methods* | | | | | | | | | | | | | | | | | | |
| **MPLight** | 407.20 | 370.90 | 5873 | 291.10 | 293.13 | 4311 | 315.45 | 319.35 | 5322 | 306.73 | 309.48 | 2930 | 361.25 | 359.70 | 5543 | 1653.93 | 545.21 | 6346 |
| **CoLight** | 360.29 | 363.24 | 6119 | 281.95 | 283.46 | 4309 | 275.57 | 278.06 | 5364 | 328.49 | 329.32 | 2923 | 349.91 | 350.05 | 5351 | 1115.76 | 526.99 | 7898 |
| **Efficient-MPLight** | 268.13 | 269.81 | 6165 | 248.95 | 250.80 | 4315 | 265.20 | 267.90 | 5373 | 285.77 | 288.35 | 2931 | 311.58 | 312.64 | 5381 | 1353.41 | 557.54 | 7572 |
| **Efficient-CoLight** | 264.72 | 266.49 | 6180 | 243.71 | 245.44 | 4314 | 258.82 | 261.37 | 5379 | 281.04 | 283.53 | 2932 | 310.79 | 311.46 | 5391 | 1284.65 | 564.87 | 7716 |
| **Efficient-MaxPressure** | 262.07 | 263.51 | 6177 | 238.67 | 240.15 | 4315 | 246.98 | 249.19 | 5389 | 274.34 | 276.43 | 2934 | 309.22 | 310.06 | 5396 | 1126.45 | 584.56 | 7891 |
| **Efficient-PressLight** | 267.24 | 268.70 | 6167 | 246.92 | 248.52 | 4314 | 253.16 | 255.41 | 5381 | 279.31 | 281.74 | 2932 | 311.12 | 311.90 | 5390 | 1182.93 | 575.23 | 7811 |
| **Advanced-MPLight** | 289.74 | 291.83 | 6221 | 247.69 | 249.36 | 4339 | 249.31 | 251.64 | 5398 | 318.62 | 318.10 | 5493 | 312.87 | 311.44 | **5546** | 1078.66 | 503.92 | 8048 |
| **Advanced-CoLight** | 283.55 | 285.22 | 6239 | 244.81 | 246.09 | 4341 | 246.78 | 248.91 | 5403 | 316.74 | 316.58 | 5512 | 316.74 | 316.58 | 5512 | 1071.43 | **500.37** | 8079 |
| **Advanced-MaxPressure** | 262.07 | 263.51 | **6260** | 238.67 | 240.15 | **4355** | 241.11 | 243.36 | **5422** | 269.62 | 271.54 | **5514** | 312.87 | 310.06 | **5546** | 1059.88 | 493.16 | 8121 |
| *Grouped RL-based methods* | | | | | | | | | | | | | | | | | | |
| **GPLight** | 496.13 | 446.81 | 5703.78 | 382.49 | 343.37 | 4213.18 | 370.76 | 344.92 | 5222.6 | 758.17 | 534.88 | 2550.93 | 679.84 | 492.23 | 4074.2 | 929.43 | 920.54 | 6261.72 |
| **SLight** | **246.14** | **248.16** | 6182 | **234.01** | **235.34** | 4316 | **230.90** | **232.81** | 5392 | 272.12 | 274.35 | 2933 | **304.11** | **305.35** | 5234 | **870.41** | 532.57 | **8642** |

*Table 1.* Comparison of traffic signal control methods across real-world datasets (JN, HZ, NY1). Each dataset is evaluated using three metrics: average travel time (ATT, lower is better), average waiting time (AWT, lower is better), and throughput (TP, higher is better). Best results are in **bold** and second-best results are underlined.

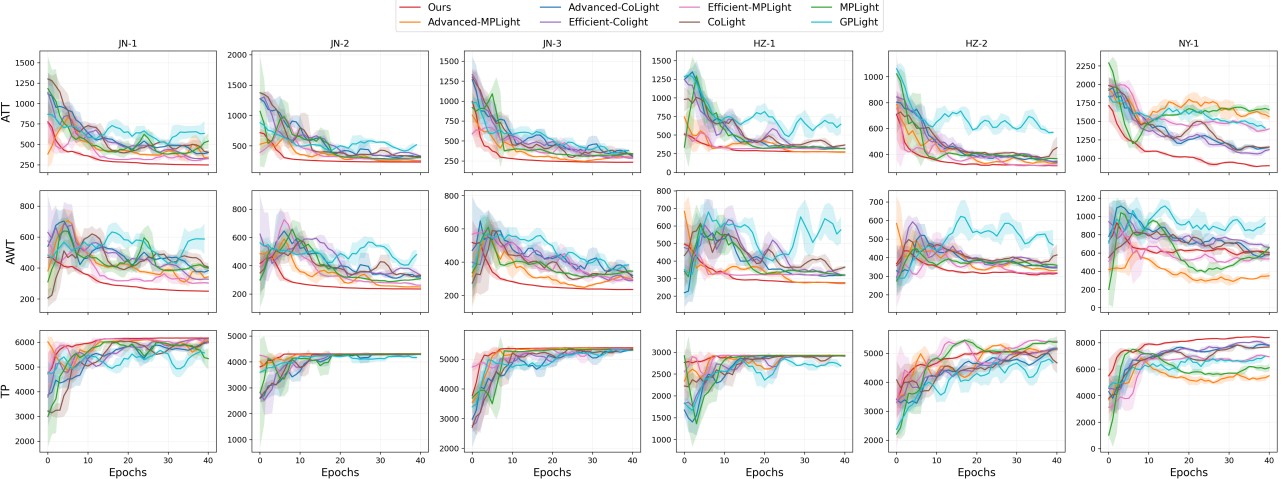

*Figure 4.* Learning curves of ATT (top), AWT (middle), and TP (bottom) over 40 training epochs on JiNan, HangZhou, and NY1 under different demand settings. Solid lines denote mean performance and shaded bands indicate variability across training.

# 5. Experiments

## 5.1. Settings

**Datasets**. We evaluate SLight on JiNan (JN), HangZhou (HZ), and New York/Manhattan (NY) benchmarks commonly used in TSC studies (*e.g.,* (Wei et al., 2019b)). JN has 12 intersections (3×4) with three demand settings containing 4,365, 5,494, and 6,295 routes; HZ has 16 intersections (4×4) with two demand settings containing 2,983 and 6,984 routes. NY1 is the standard 196-intersection Manhattan benchmark (28×7) with 16,337 taxi-derived routes, and NY2 is a larger Manhattan network with 2,510 intersections used to stress-test metropolitan-scale scalability. These settings separate two evaluation goals: JN/HZ test

robustness under demand variation on small regular grids, while NY1/NY2 test whether shared-policy routing remains effective as network heterogeneity and scale increase.

**Compared Baselines**. We compare SLight against representative **rule-based** and **learning-based** methods under the CityFlow (Zhang et al., 2019) interface. Traditional baselines include **FixedTime** (Koonce et al., 2008) and **Max-Pressure** (Varaiya, 2013). Unstructured RL-based baselines include **MPLight** (Chen et al., 2020), **CoLight** (Wei et al., 2019b), and **PressLight** (Wei et al., 2019a), together with their efficiency/representation-enhanced variants (**Efficient-** (Wu et al., 2021) and **Advanced-** (Zhang et al., 2022)). We also include **GPLight** (Liu et al., 2023) as a grouped RL-based baseline to test whether grouping intersections

by learned traffic representations is sufficient for scalable coordination.

**Evaluation Metrics and Protocol**. We report **Average Travel Time (ATT)**, **Average Waiting Time (AWT)**, and **ThroughPut (TP)** (Table 1); ATT is the primary metric (Wei et al., 2019b). RL-based methods are trained for 40 episodes (each episode is a 60-minute simulation), and results are reported as the mean/variance over the last ten evaluation episodes unless stated otherwise.

**Implementation and Environment**. Experiments run on a server with two Xeon Gold 8375C CPUs (32C/64T) and 2TB RAM; learning-based methods use an NVIDIA RTX 4090 (24GB). All learning-based methods share the same simulator interface and control interval $\Delta t$ (fixed-phase), consistent with Sec. 3.1–3.2.

## 5.2. Overall Performance

Table 1 shows that SLight is most beneficial when coordination difficulty grows from small grids to the heterogeneous Manhattan network. On JN and HZ, several pressure-aware baselines are already strong because these regular grids have limited structural diversity; SLight still obtains the best or near-best ATT by adapting shared policies to demand shifts rather than relying on a fixed grouping. On NY1, the gap becomes larger: SLight reduces ATT while also achieving the highest TP, suggesting that the main scalability bottleneck is not policy sharing itself but whether each intersection is routed to a policy whose behavior matches its evolving traffic pattern. GPLight's weaker results support this interpretation, as grouping by traffic representation alone does not reliably produce behaviorally compatible shared policies.

Table 2 extends this observation to NY2 with 2,510 intersections. SLight reduces ATT to 856.94, outperforming both rule-based controllers and representative RL-based baselines on the larger Manhattan network. The result indicates that a small set of learned principles can absorb substantial network heterogeneity when assignment is updated by policy-aware traffic queries, making the framework scale as a routing-and-sharing design rather than as independent per-intersection learning.

## 5.3. Interpreting Shared Principles

We further examine whether the learned shared principles are merely latent routing artifacts or whether they expose meaningful traffic-control structure. Since the principles are learned without semantic supervision, we interpret them through two observable properties: whether each principle is selected often enough to remain active, and whether its prototype traffic fingerprint corresponds to a coherent control regime. Table 3 summarizes these properties on NY1

by reporting the usage of each principle together with its representative queue, pressure, directional imbalance, and resulting traffic-regime interpretation.

The usage ratios are close to uniform, ranging from 22.98% to 26.61%, with raw assignment counts between 901 and 1043. This rules out two uninformative outcomes: SLight does not collapse to a single shared policy, and it also does not leave some principles unused. The policy pool therefore remains active, supporting the use of a small set of shared principles rather than one global controller or many independent controllers.

The prototype fingerprints further show that the principles are not separated only by their selection frequency. Principle 1 has the largest north–south/east–west queue values, 0.41/0.28, and the largest pressure values, 0.52/0.37, with positive across-principle $z$-scores in both directions. Its imbalance entry, 0.65/0.13, also indicates both high directional imbalance and a positive north–south minus east–west signal. These values justify its interpretation as a heavy north–south-dominant regime. Principle 2 has lower queue and pressure values, especially in the east–west direction, as shown by the negative east–west $z$-scores in both queue and pressure. However, its signed north–south minus east–west signal remains positive, so it corresponds to a lighter but still north–south-biased regime. Principle 3 lies between these two cases: its east–west queue and pressure are near or slightly above the across-principle average, while its imbalance is moderate. This makes it closer to a medium mixed-load regime. Principle 4 has the lowest north–south queue and the smallest signed north–south minus east–west signal among the four principles, so it is the closest to a balanced regime.

These distinctions indicate that the learned principles capture both traffic intensity and directional structure. This is important for traffic signal control because phase selection depends not only on how congested an intersection is, but also on which movement direction requires priority. The traffic-regime labels in Table 3 are therefore not separate supervision signals; they are post-hoc summaries of the queue, pressure, and imbalance fingerprints.

## 5.4. Learning Efficiency and Stability

Figure 4 reports learning curves of ATT, AWT, and TP over 40 training epochs. Across JiNan, HangZhou, and NY1, SLight reduces ATT and AWT more rapidly than the baselines and reaches a stable regime in fewer epochs. The narrower shaded bands indicate lower run-to-run variability, which suggests that adaptive assignment reduces the instability caused by repeatedly training shared policies on incompatible traffic patterns. On NY1, where multi-agent coupling is strongest, SLight continues improving ATT while maintaining high TP, showing that the routing

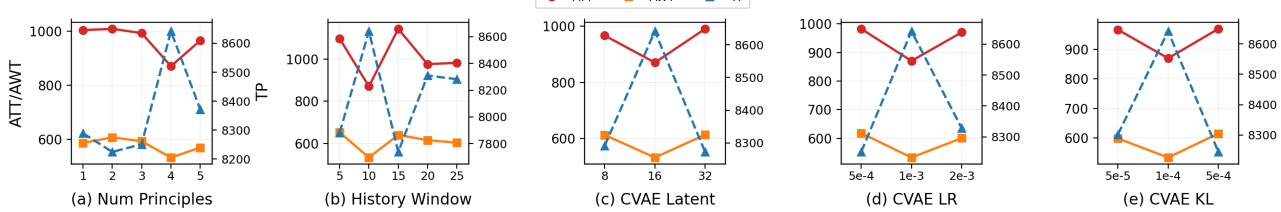

*Figure 5.* Hyperparameter sensitivity on NY1 (196 intersections). We vary (a) number of principles $K$, (b) LSTM history window length $\tau$, (c) CVAE latent dimension, (d) CVAE learning rate, and (e) CVAE KL weight.

*Table 2.* Metropolitan-scale evaluation on NY2, the Manhattan 2.5K network, using ATT.

| Method | NY2 ATT ↓ |
|---|---|
| Fixed-Time | 1844.56 |
| MaxPressure | 1441.23 |
| Advanced-CoLight | 1138.66 |
| Advanced-MPLight | 1374.73 |
| CoSlight | 1053.90 |
| **SLight** | **856.94** |

*Table 3.* Usage and prototype traffic fingerprints of learned shared principles on NY1. Usage reports the selection ratio with the raw assignment count in parentheses. Feature entries report normalized values with across-principle $z$-scores in parentheses.

| Principle | Usage | Queue ns/ew | Pressure ns/ew | Imb. / ns-ew | Traffic regime |
|---|---|---|---|---|---|
| 1 | 26.61% (1043) | 0.41/0.28 (+1.7/+1.1) | 0.52/0.37 (+1.7/+1.1) | 0.65/0.13 (+1.6/+1.0) | Heavy NS-dominant |
| 2 | 22.98% (901) | 0.33/0.21 (-0.4/-1.6) | 0.42/0.29 (-0.6/-1.6) | 0.50/0.12 (-1.1/+0.8) | Light NS-biased |
| 3 | 25.05% (982) | 0.34/0.26 (-0.3/+0.2) | 0.43/0.34 (-0.3/+0.3) | 0.56/0.08 (-0.1/-0.4) | Medium mixed load |
| 4 | 25.36% (994) | 0.31/0.26 (-1.0/+0.4) | 0.41/0.35 (-0.8/+0.3) | 0.54/0.05 (-0.4/-1.4) | Balanced regime |

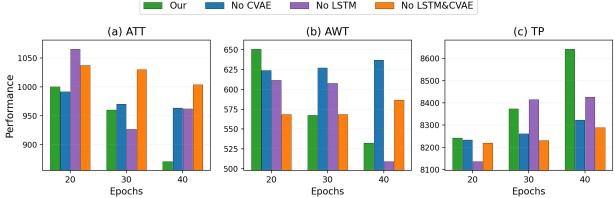

*Figure 6.* Ablation on NY1 (196 intersections) at epochs 20/30/40. (a) ATT (lower), (b) AWT (lower), (c) TP (higher).

layer helps convert shared capacity into stable network-level learning.

### 5.5. Ablation Study

Figure 6 presents the ablation study on NY1 with three variants: No CVAE, No LSTM, and No LSTM&CVAE. Removing either component degrades final ATT and TP, while removing both performs worst. Temporal encoding and principle learning solve different parts of the sharing problem: the LSTM captures delayed policy-induced traffic evolution, whereas the CVAE organizes replayed behavior into a more stable principle space. The full model therefore improves coordination not by adding redundant modules, but by linking temporal traffic queries with reusable policy prototypes.

The AWT results should be interpreted together with ATT and TP. In some checkpoints, an ablated variant can obtain competitive AWT while producing lower TP, which reflects a changed set of completed trips rather than better network-level control. The full SLight model achieves the best overall balance across ATT, AWT, and TP.

### 5.6. Hyperparameter Analysis

Figure 5 reports hyperparameter sensitivity on NY1 using ATT/AWT (lower) and TP (higher). The trends show that SLight benefits from moderate routing capacity rather than maximal model complexity. A small $K$ underfits heterogeneous intersection roles, while an overly large $K$ fragments data across too many shared policies; $K{=}4$ gives the best overall balance and is used by default. Similarly, $\tau{=}10$ provides the best ATT–TP trade-off because shorter histories miss delayed control effects and longer histories can introduce stale dynamics. The CVAE latent dimension, learning rate, and KL weight indicate that stable principle learning requires sufficient but not excessive latent capacity and regularization.

## 6. Conclusion

We propose *SLight*, a scalable shared-policy framework for multi-agent traffic signal control that decouples reusable policy learning from non-stationary traffic patterns. By encoding each intersection with a policy-aware traffic pattern embedding and adaptively assigning it to one of $K$ learned *control principles*, SLight enables efficient sharing under heterogeneity and improves network-level efficiency at scale. *SLight* currently belongs to simulator-based optimization within a fixed road network and fixed control interval; extending it to cross-city transfer, variable green-time allocation, broader heterogeneous networks remains future work.

## Impact Statement

This work studies adaptive traffic signal control, which can affect urban mobility, safety, and routing services. SLight is evaluated in traffic simulators and is not intended for direct real-world deployment without municipal validation, safety constraints, and traffic-engineering oversight. In deployment, dynamic signal policies should also be monitored for their interactions with navigation systems that rely on stable signal timing for ETA estimation and route planning.

## Acknowledgment

This work was supported by National Key R&D Program of China under Grant #2024YFA1012700, National Natural Science Foundation of China #62572414, and Guangdong-Hong Kong Technology Innovation Joint Funding Scheme #2024A0505040012.

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

*Table 4.* Additional ATT-only comparison with recent RL-based and LLM-based TSC baselines. Lower is better.

| Model | JN1 | JN2 | JN3 | HZ1 | HZ2 | NY1 |
|---|---|---|---|---|---|---|
| CoSLight | 391.63 | 269.93 | 289.40 | 389.30 | 441.66 | 1404.99 |
| TransformerLight | 256.16 | 235.78 | 232.39 | 273.18 | 313.91 | 1097.33 |
| Proactive-CoLight | 431.07 | 435.14 | 416.86 | 433.16 | 470.64 | 1024.62 |
| GPLight+ | 329.14 | 278.94 | 287.10 | 312.99 | 410.19 | 1181.42 |
| Llama3.3-70B | 272.41 | 244.55 | 243.53 | 281.44 | 326.42 | – |
| Qwen2.5-72B | 275.42 | 251.41 | 264.21 | 282.13 | 329.34 | – |
| GPT-3.5-turbo | 337.32 | 328.19 | 343.19 | 293.42 | 348.59 | – |
| GPT-4o | 281.58 | 259.61 | 258.85 | 280.48 | 325.48 | – |
| DeepSeek-R1-671B | 279.11 | 258.43 | 262.21 | 278.56 | 335.53 | – |
| DeepSeek-R1-Distill-7B | 331.45 | 311.43 | 288.42 | 291.32 | 344.73 | – |
| Traffic-R1-3B | 270.34 | 239.53 | 238.03 | 277.83 | 324.11 | – |
| *SLight* | **246.14** | **234.01** | **230.90** | **272.12** | **304.11** | **870.41** |

# A. Additional Experimental Details

This appendix provides supplementary experiments that support the main empirical claims without changing the structure of the primary comparison table. The main text reports the complete ATT/AWT/TP comparison and focuses on the new metropolitan-scale and principle-interpretation analyses. Here we report additional ATT-only baselines, a small-network principle-capacity check, routing alternatives, and hyperparameter interpretations.

## A.1. Additional ATT-only Baseline Comparisons

Table 4 compares *SLight* with recent RL-based and LLM-based TSC baselines under average travel time (ATT). We keep this comparison separate from the main table because these additional baselines provide ATT only, whereas the main table jointly reports ATT, AWT, and TP. This separation keeps the main benchmark internally consistent while still documenting the broader baseline coverage requested during review.

The main observation is consistency rather than a single isolated win. Among the additional baselines, TransformerLight is the strongest competitor on most JiNan and HangZhou settings, but *SLight* still achieves the lowest ATT on all shared settings. The margins are small on JN2, JN3, and HZ1, which suggests that these smaller networks can already be handled reasonably well by strong sequence-modeling or LLM-based controllers. The advantage becomes clearer on HZ2 and especially NY1, where *SLight* reduces ATT to 870.41, substantially below the best additional baseline on NY1. This trend is consistent with the paper's main claim: policy-aware routing is most valuable when network scale and intersection heterogeneity amplify the cost of mismatched sharing.

LLM-based baselines are evaluated on JiNan and HangZhou, where full-shot reproduction is feasible. NY-scale LLM results are not reported because public training pipelines and model-specific reproduction settings are unavailable, and the computational cost is high for large-network full-shot training.

## A.2. Additional Principle Analysis

The main text shows that the learned principles on NY1 are all used and correspond to different traffic regimes. Table 5 tests a complementary question: whether multiple principles are also useful on a smaller network. We vary the number of principles on JN-1 and report ATT.

The result shows a monotonic reduction in ATT as the number of principles increases from one to four. A single principle already provides a shared-policy controller, but it cannot represent the distinct local roles and demand variations that appear even in a small network. Moving from one to four principles reduces ATT from 264.72 to 246.14, a relative reduction of about 7.0%. The improvement from three to four principles is smaller than the earlier gains, suggesting that a small policy pool is sufficient; the benefit comes from a modest number of reusable control regimes rather than from assigning every intersection its own policy.

*Table 5.* Effect of the number of shared principles on JN-1, measured by ATT. Lower is better.

| Number of principles | 1 | 2 | 3 | 4 |
|---|---|---|---|---|
| JN-1 ATT | 264.72 | 260.43 | 248.32 | **246.14** |

*Table 6.* Routing alternatives on NY1 measured by ATT. Lower is better.

| Routing design | NY1 ATT |
|---|---|
| Direct clustering of recurrent embeddings | 929.43 |
| MoE-style routing | 963.44 |
| CVAE-based principles / *SLight* | **870.41** |

## A.3. Additional Ablation and Routing Alternatives

The main ablation evaluates whether the LSTM encoder and CVAE principle learner are useful when embedded in the full *SLight* pipeline. Table 6 asks a more specific design question: whether the CVAE-based principle representation is necessary, or whether simpler routing mechanisms can provide the same benefit.

Direct clustering of recurrent embeddings represents a similarity-driven grouping strategy: intersections are grouped according to learned traffic representations, and policy assignment follows the induced clusters. MoE-style routing tests whether a generic gating module is sufficient once multiple shared policies are available. Both alternatives underperform the CVAE-based principle representation. *SLight* improves ATT by about 6.4% over direct clustering and about 9.7% over MoE-style routing on NY1. The takeaway is that the gain does not come merely from adding a routing branch. The benefit comes from learning behavior-grounded shared principles and routing intersections by policy suitability rather than by traffic similarity alone.

## A.4. Additional Hyperparameter Analysis

The hyperparameter study in the main text is intended to identify robust operating regions rather than to tune each dataset independently. Across the sweeps, the same pattern appears: overly small settings underfit heterogeneous traffic roles, while overly large or weakly regularized settings fragment experience and make assignment less stable under non-stationary MARL training.

For the number of principles, small $K$ limits specialization because too many intersections must share the same policy behavior. Large $K$ has the opposite problem: each policy receives less experience, and the assignment module has more opportunities to fragment traffic patterns into weakly trained groups. The best setting near $K = 4$ balances these effects by providing enough policy diversity without losing the statistical efficiency of sharing.

The history window length follows a similar trade-off. A short window may miss delayed effects of phase decisions, such as queue dissipation and spillback propagation. A very long window can include stale dynamics generated by older policies, which is undesirable because the traffic-pattern embedding is used for assignment during ongoing learning. An intermediate window therefore provides the most useful evidence for policy-aware routing.

For the CVAE, moderate latent dimension, learning rate, and KL weight are preferable. The latent space must be expressive enough to separate traffic regimes, but not so unconstrained that prototypes drift with each policy update. In this sense, the hyperparameter results support the same interpretation as the routing-alternative study: *SLight* works best when the principle space is compact, behavior-grounded, and stable enough to serve as a reusable policy index.

