# OpenReview forum: "Scalable Traffic Signal Control with Shared Policy Framework"
_ICML.cc/2026/Conference — ICML 2026 regular_

### Official Review · Reviewer_Bgwy · 2026-02-17

**Soundness:** 3
**Presentation:** 3
**Significance:** 3
**Originality:** 2
**Overall Recommendation:** 4
**Confidence:** 4

**Summary:**

This paper proposes SLight, a multi-agent reinforcement learning framework for scalable traffic signal control. SLight decouples the grouping of control policies from the grouping of intersection traffic patterns through three components: (1) a policy-aware traffic pattern encoder that uses a long short-term memory recurrent network to encode windows of observation, action, and reward tuples into traffic pattern embeddings; (2) a shared control principle representation that trains a conditional variational autoencoder on state-action interactions to learn a fixed number of latent control principles, maintained as prototypes via momentum averaging; and (3) an adaptive assignment module that uses a bilinear compatibility score to dynamically assign each intersection to one of the shared policies. Experiments are conducted on three CityFlow benchmarks (JiNan, HangZhou, New York), and SLight reports the best average travel time across settings, with the most pronounced gains on the 196-intersection New York network.

**Compliance With Llm Reviewing Policy:**

Affirmed.

**Key Questions For Authors:**

1. Why is the largest evaluation network (196 intersections) smaller than GPLight's own evaluation (1,089 intersections)? Since scalability is the paper's central claim, can you provide results on networks of 500+ and 1,000+ intersections? A positive answer with results on 1,000+ intersection networks would significantly strengthen the paper and could raise my score.

**Limitations:**

The paper does not adequately discuss several key limitations: (i) evaluation scale is much smaller than the metropolitan-scale motivation, (ii) the fixed-phase control interval simplifies the problem relative to real-world variable green time allocation, (iii) no reporting of compute overhead from the added LSTM, CVAE, and prototype maintenance, and (iv) no transfer experiments across cities or to unseen intersections.

**Strengths And Weaknesses:**

Strengths

S1. The paper targets a real and widely observed failure mode in shared policy traffic control: as learning progresses, the representation of "similar" intersections drifts, and similarity-based grouping can cause unstable reassignment and destructive interference. By treating the traffic embedding as a drifting query and the shared principles as a slower, behavior-grounded reference, the paper provides a clean decomposition.

S2. The method is modular and reasonably well-motivated. The policy-aware encoder incorporates both the previous action and its outcome (reward), which is a sensible step beyond purely state-based grouping. The principle learning objective aligns with capturing decision tendencies rather than snapshot traffic states, which is exactly what parameter sharing needs to preserve.

Weaknesses

W1. The scalability claim is not adequately supported. The paper positions scalability as a central contribution, yet the largest evaluation is 196 intersections. Prior work has evaluated grouped or decentralized reinforcement learning traffic signal control at substantially larger scales, including GPLight [1] on 1,089 intersections and MPLight [2] on 2,510 Manhattan traffic lights. As a result, it is unclear whether the proposed approach holds at scale.

W2. The compatibility score conditions on a one-hot intersection identity feature, which ties the learned principles to specific intersections. This makes it unclear whether the framework can transfer to unseen intersections or new cities without retraining.

W3. Presentation is harder to follow on pages 6, 7, and 8 due to over-formatting. There are many bolded phrases (sometimes multiple per sentence) and dense result narration that makes it difficult to parse the key claims. A representative example is the following sentence, which feels overly compressed and visually noisy (verbatim from the paper): "Fig. 6 ablates SLight on NY (196 intersections) with No CVAE, No LSTM (no history encoder), and No LSTM&CVAE, reporting ATT/AWT/TP at epochs 20/30/40. All components help."


[1] GPLight: Grouped Multi-agent Reinforcement Learning for Large-scale Traffic Signal Control. Yilin Liu et. al.  IJCAI 2023.

[2] Toward A Thousand Lights: Decentralized Deep Reinforcement Learning for Large-Scale Traffic Signal Control. Chacha Chen et. al. AAAI 2020

---

> ### Author Rebuttal · Authors · 2026-03-31
>
> We sincerely thank the reviewer for the detailed and constructive comments. We especially appreciate your recognition of the paper’s motivation, the clean decomposition of shared principles and traffic patterns, and the well-motivated policy-aware design. We hope our response addresses your concerns, and we are happy to clarify any remaining questions.
>
> ### **[W1&Q1] Scalability**
> Thank you for your insightful comment.
>
> We agree that the original evaluation up to 196 intersections was insufficient for a central scalability claim. We have now added results on the Manhattan network with 2,510 intersections, which directly extends the evaluation to the 1,000+ scale raised by the reviewer.
>
> As shown in the added table, SLight achieves best average travel time (ATT) on this larger network against representative rule-based and RL-based baselines. Due to rebuttal-time constraints, we currently report ATT as the end-to-end metric on Manhattan.
> | **Model** | **Manhattan** |
> | --- | --- |
> | Fixed-Time | 1844.56 |
> | MaxPressure | 1441.23 |
> | Advanced-Colight | 1138.66 |
> | Advanced-Mplight | 1374.73 |
> | CoSlight | 1053.90 |
> | **Slight** | **856.94** |
> ### **[W2] Transferability**
> Thank you for your comment.
>
> We would like to clarify that the current framework is not intended to transfer to unseen intersections or new cities without retraining, but to optimize specifically for each network to identify its most suitable policies. The one-hot identity feature is used only to stabilize assignment within a fixed network.
>
> Our scalability claim is therefore about handling large heterogeneous networks within a given network instance, not about cross-network transfer. The shared control principles are still intended to capture shared behavioral patterns from policy-aware interactions rather than to act as location-specific lookup tables. We will revise the paper to make this boundary explicit.
>
> ### **[W3] Presentations**
> Thank you for pointing this out. We agree that pages 6–8 are currently harder to follow than necessary. In the revision, we will reduce bold emphasis to only the main takeaways and replace compressed result narration with shorter declarative sentences.
>
> For the specific sentence you highlighted, we will rewrite such sentences in a cleaner form:
>
> Figure 6 presents the ablation study on NY (196 intersections). We compare SLight with three variants: No CVAE, No LSTM, and No LSTM&CVAE. Results at epochs 20, 30, and 40 show that removing either module consistently degrades ATT, AWT, and TP, indicating that both components contribute to performance.
>
> Due to rebuttal space, we cannot paste the full rewritten pages here, but this will be a structural rewrite rather than minor wording edits.
> ### **Limitation**
> We agree that these limitations should be stated more explicitly.
>
> (i) We agree that the original evaluation scale was smaller than the metropolitan-scale motivation. We have now added results on the 2,510-intersection Manhattan network to better support the scalability claim.
>
> (ii) We acknowledge this limitation. Our current formulation assumes a fixed-phase control interval, which is a simplifying assumption compared with real-world settings that allow variable green-time allocation. We will make this scope explicit in the revision and discuss variable-duration control as an important direction for future work.
>
> (iii) We agree that the original submission did not report the extra computational cost introduced by the LSTM, CVAE, and prototype maintenance.
>  - We have now added computational-overhead analysis [e.g., inference time and parameter count]. The CVAE and LSTM introduce only marginal inference overhead with better performance.
> | Method | Time (s) | Params |
> | --- | --- | --- |
> | SLight | 0.018824 | 156,936 |
> | No CVAE | 0.017550 | 108,260 |
> | No LSTM | 0.018597 | 72,520 |
> | No LSTM&CVAE | 0.009580 | 23,844 |
> - We also analyze the **principle distribution** in table and the **traffic patterns** associated with each policy in NY, as illustrated in **Figure 1** of the anonymous repository (`https://anonymous.4open.science/r/SlightRebuttal-CC66/`).
> | Principle ID | Decision count | Global usage ratio |
> | --- | --- | --- |
> | 0 | 1043 | 26.61% |
> | 1 | 901 | 22.98% |
> | 2 | 982 | 25.05% |
> | 3 | 994 | 25.36% |
> - **Policy 0:** heaviest traffic, strongest north-south bias.
> - **Policy 1:** light traffic, still north-south dominant.
> - **Policy 2:** medium-to-heavy traffic with moderate imbalance.
> - **Policy 3:** most balanced traffic regime.
>
> Detailed analysis is provided in **Reviewer  mihN [W2]**.
>
> (iv) The current framework is not designed for cross-city or unseen-intersection transfer without retraining. It is designed for stable assignment within a fixed network, rather than identity-invariant transfer across different networks. We will revise the paper to make this boundary explicit and avoid overstating generalization.

---

> > ### Author Rebuttal · Reviewer_Bgwy · 2026-03-31
> >
> > My concerns are resolved. After carefully considering all the comments and responses, I have revised the score from weak reject to weak accept.

---

> > > ### Author Response · Authors · 2026-04-03
> > >
> > > Thank you very much for your careful reconsideration and for revising the score from weak reject to weak accept. We sincerely appreciate your thoughtful evaluation and are very glad that our responses have adequately addressed your concerns.

---

### Official Review · Reviewer_hs9V · 2026-03-06

**Soundness:** 2
**Presentation:** 2
**Significance:** 2
**Originality:** 2
**Overall Recommendation:** 3
**Confidence:** 5

**Summary:**

The paper proposes a MARL framework for traffic signal control where they learn a recurrent embedding of each intersection’s recent traffic history, match that embedding to latent group prototypes, and then select among a small number of shared policies for each intersection.

**Compliance With Llm Reviewing Policy:**

Affirmed.

**Final Justification:**

The authors have provided additional experiments and other details that partially address my concerns. I also understand that this is not a theoretical contribution and am not evaluating it in that light at all; rather, I am not fully convinced that the novelty of the framework is coming across, at least as stated in its current form, without sounding like a linear combination of several approaches. With these concerns and given the authors' responses, I have raised my score to 3.

**Key Questions For Authors:**

See weaknesses above.

**Limitations:**

No limitations of the proposed approach are discussed. Please discuss possible limitations (complexity, robustness, etc.) of the proposed approach.

**Strengths And Weaknesses:**

Strengths:
1. Extensive benchmarking against rule-based and RL-based methods.
2. Scales to large networks and overall, strong overall results in NY dataset

Weaknesses:
1. Regarding novelty:
(a) I don’t understand what is novel here beyond a combination of an LSTM encoder to embed shared policy history with a standard MARL traffic signal control setup, to select among K control policies.
(b) There is an excessive amount of jargon such as “policy-aware”, “control principles”, and “adaptive assignment”, none of which is defined or standard in the field. In my understanding, “policy sharing” just means multiple intersections reuse one of (K) shared policies, and “adaptive assignment” just means choosing which shared policy each intersection uses at a given time, and “control principles” seem to just be latent group prototypes. I would rather the authors state this more plainly rather than rebrand standard routing/clustering ideas into vague RL-sounding terms.
(c) Specifically, the main claim to novelty is “policy-aware” RL. But isn’t all RL policy aware? All observed state trajectories already depend on past actions and the current policy, and replay buffers already store state-action-reward transitions. It is not clear what the RL contribution is here.
(d) More bluntly, I am not convinced that the paper distinguishes “better representation for routing” from “better RL algorithm.” Most of the method looks like a routing layer on top of a standard Q network. While that may still be useful in this application domain, but the contribution should be described at that level.
(e) The authors should tone down claims like grouping based on embeddings is “irrelevant”. If the problem is that the embedding drifts during learning, then that is a very common issue across all nonstationary representation learning, not just an issue with prior grouped RL based traffic control approaches.
2. The authors claim their approach is more “interpretable” in the contribution list (line 80, right column). However, there is no further mention of interpretability anywhere else afterwards, nor is there any insight on what these shared so-called “control principles” are supposed to mean in the traffic sense (e.g., favor arterial flow, prioritize side street, etc.), and why they work better and/or are more reusable/reinterpretable than existing policies? Thus, the claim of interpretability is invalid here.
3. There is a fundamental issue of complexity vs benefit here. The proposed method uses an LSTM encoder, a CVAE, prototype maintenance, replay-based prototype updates, a separate assignment module, and multiple shared Q-networks. I am not yet convinced that all this extra machinery is necessary or justified by sufficient evidence that simpler shared policies cannot work reasonably well in this setting without this added complexity.
4. The routing decision depends directly on a one-hot intersection identity $id(i)$ (lines 293-295, right column). Essentially, some of the heterogeneity is hard coded rather than discovered through shared behavioral structure. This weakens the claim that the learned groups reflect reusable “control principles” rather than location-specific lookups.
5. Experiments:
(a) The need for CVAE or the advantage of the variational structure here is not clear. The ablation only compares against “No CVAE”; a comparison at least with respect to simpler alternatives like direct clustering of recurrent embeddings or a standard mixture-of-experts routing is needed to validate the benefit of using CVAE.
(b) The ablation is narrow and only on NY at epochs 20, 30, and 40. That is not sufficient to justify the fairly elaborate claims about stability and scalability of the proposed method.
(c) The paper claims CVAE gives a more stable latent principle space, but there are no experimental results to back up this claim (e.g., statistics on assignment switching, action variance, etc.)

---

> ### Author Rebuttal · Authors · 2026-03-31
>
> Thanks for the thoughtful feedback.
>
> **W1**
>
> a.  SLight is not a trivial combination, but a novel **routing-and-sharing framework** built on shared-policy MARL for TSC. Different from previous SOTA GPLight (group intersections on learned traffic representations), SLight separates traffic-pattern grouping from policy grouping by learning shared control principles and using traffic embeddings only for adaptive routing. Experiments show that SLight performs best by identifying and switching to more suitable policies flexibly.
>
> b. Yes, your understandings are correct. These concepts were defined in the paper (policy-aware in Fig-1c and line 105 left; control principles in lines 178–179 right; and assignment in line 206 left), but they are in different locations, and some are late. We will define them earlier with plainer language (policy-aware encoder: assignment representation from recent observations and actions; control principles: shared policy prototypes; adaptive assignment: routing each intersection to one shared policy).
>
> c. It is a misunderstanding. Yes, all RL have policy, but due to the decoupling of policy and intersection, our “policy-aware” means the traffic pattern matching in the policy assignment: when the traffic condition of an intersection changes, we identify the most suitable policy for it, not like the existing ones that stick to the same policy.
>
> d. Yes, “better representation for routing” is the right level for SLight as it is a routing-and-sharing framework rather than a new RL algorithm.
>
> e. We will soften the wording around “irrelevant” embeddings. Our intended claim is narrower: under non-stationary learning, directly grouping on drifting embeddings can cause a mismatch between traffic similarity and policy suitability.
>
> **W2**
>
> We omitted it due to page limit, and how we have provided 3 new analysis:
> 1) We output the utility percentage of the 4 learned shared policies in NY, and they are all meaningfully used rather than collapsing to 1 dominant mode: 26.61%, 22.98%, 25.05%, and 25.36% global usage.
>
> 2) The learned principles correspond to distinct traffic regimes, and their traffic patterns are illustrated in Fig1 of repository: `https://anonymous.4open.science/r/SlightRebuttal-CC66/`.  Specifically, one policy captures the heaviest north-south-dominant traffic, one captures light traffic, one corresponds to medium-to-heavy traffic with moderate directional imbalance, and one is the most balanced regime. Detailed analysis is provided in **mihN [W2]**.
>
> 3) The per-intersection usage heatmap in Fig3 of repository shows clear spatial specialization in NY: different intersections keep switching to different shared policies rather than sticking to one single policy.
>
> **W3**
>
> We agree that the complexity-benefit tradeoff could be more explicitly.
>
> Our ablation (Fig 6) verifies the necessity of each component: removing LSTM encoder or CVAE degrades performance, and removing both performs worst.
>
> We have also added computational-overhead analysis, please see **Bgwy [Limitation] (iii)**. In brief, the CVAE and LSTM introduce less than 0.01s while achieving best performance.
>
> We compare with direct clustering of recurrent embeddings and MoE routing. In NY, both are worse than CVAE-principle learning (ATT: clustering 929.43, MOE 963.44, CVAE 870.41), suggesting that the gain is not from adding an arbitrary routing branch, but from explicitly learning shared latent principles.
>
> **W4**
>
> The one-hot intersection identity is used only in the assignment module to stabilize routing within a fixed network. The current framework is therefore not designed for identity-invariant transfer to unseen intersections or new cities without retraining. Our scalability claim is about handling large heterogeneous networks within one network instance, not cross-city transfer.
>
> **W5**
>
> Thank you for this detailed comment. We address each point below.
>
> For the concern on CVAE and stability, please also see **W2** and **W3** above. In brief, for **(a)**, CVAE outperforms simpler routing alternatives; for **(b)**, we have added training curves in Fig2 to provide broader evidence beyond the original narrow ablation; and for **(c)**, the added policy-distribution analysis, prototype visualization Fig1, and per-intersection usage heatmap Fig3 provide more direct evidence that the learned shared policies are meaningful and more stable in practice: `https://anonymous.4open.science/r/SlightRebuttal-CC66/`.
>
> For broader concerns on scalability and stronger baselines, please see **Bgwy [W1]** and **mky5/mihN [newer baselines]**. In brief, we have added Manhattan results with 2510 intersections, where SLight achieves the best ATT among representative baselines, and we have also added more recent RL/LLM-based baselines.
>
> **Limitations**
> We will discuss the limitations of generalizability and transferability of SLight as it aims to optimize for each network specifically.

---

> > ### Author Rebuttal · Reviewer_hs9V · 2026-04-01
> >
> > Thanks for the response and additional experiments.
> >
> > While they address some of my questions, my main issue regarding novelty remains. I specifically asked why this is more than a combination of a recurrent history encoder, a latent routing or grouping module, and a small set of shared policies, but the rebuttal mainly repeats the same intuition in different words rather than identifying clearly what the new technical idea is here. I appreciate the clarification that this is a routing-and-sharing framework rather than a new RL algorithm. But that directly reinforces my original concern that the method looks like a routing layer on top of standard shared-policy MARL, and the rebuttal still does not explain what is new beyond that. The “policy-aware” claim also remains weakly supported. As written, it seems to mean only that the routing representation includes recent observations, actions, and rewards. While this is a reasonable design choice, it is not by itself a strong contribution.
> >
> > Second, I appreciate the clarification on the one-hot intersection identity being used only within a fixed network and not for transfer. But that does not resolve the concern. Once the identity is given directly to the assignment module, the method can rely in part on a location-specific information rather than discovering shared behavioral structure. That makes it unclear whether the learned groups reflect genuine shared control principles at all, or whether they are partly just location-specific lookup decisions tied to intersection identity.
> >
> > Based on these concerns, I maintain my score.

---

> > > ### Author Response · Authors · 2026-04-03
> > >
> > > We sincerely appreciate the reviewer’s time and careful feedback. Thanks for helping us summarize and clarify the technical idea here. Yes, conceptually on the highest level, this work follows the combination you mentioned. However, from the perspective of TSC research, we were not simply apply this idea to TSC problem, but happens to come up with this similar general RL idea again (that is reason why we kept emphasizing on the policy-aware and routing-and-sharing framework…). In fact, the novelty stays in the line of TSC research itself. In this field, the RL was introduced and utilized incrementally, from Jessie’s CoLight that formulates TSC as an RL problem, to the newest GPLight that cluster signals into groups and learn policy for each group. Our research standing upon GPLight by breaking its fixed signal groups. But without groups, there is no stable policy, so we have to train the “principle policies” first and then map the signals to the policies flexibly.
> > >
> > > In addition, the novelty is not the routing layer itself, but the explicit latent principle space over which adaptive policy assignment is performed. A straightforward aforementioned combination embeds recent traffic, groups by that embedding, and assigns one of (K) shared policies. Its limitation is that routing stays in a **traffic-similarity space**, implicitly assuming similar traffic patterns should share the same policy. This combination is equivalent to **CROSS** [1], and we have conducted new experiments to supports the practical benefit of routing by **policy suitability** rather than direct traffic-pattern grouping.
> > >
> > > | Dataset | CROSS (Combination) | SLight |
> > > | --- | --- | --- |
> > > | JN-1 | 314.27 | 246.14 |
> > > | JN-2 | 309.88 | 234.01 |
> > > | JN-3 | 303.39 | 230.90 |
> > > | HZ-1 | 356.01 | 272.12 |
> > > | HZ-2 | 356.02 | 304.11 |
> > >
> > > [1] Chen X, Zhang Y, Xiao Y, et al. CROSS: A Mixture-of-Experts Reinforcement Learning Framework for Generalizable Large-Scale Traffic Signal Control[J]. arXiv preprint arXiv:2603.24930, 2026.
> > >
> > > Although this may still not be a significant contribution in the theoretical RL, it is a big advance in TSC such that signals can always find the most suitable policy. As evidenced by the experiment results, the performance of SLight is much better than the SOTA TSC methods. It could be viewed as an evidence this combination is very powerful and can improve TSC effectively.
> > >
> > > In fact, it is unfair for a TSC paper to have sufficient contribution in theoretical RL, as the RL research does not care (or aware of) our transportation’s TSC problem, then there could never be any advance in TSC. This can be evidenced by the following milestones of TSC research and their contributions:
> > >
> > > - **LLMLight KDD’25 Audience Appreciation Award:** introduce LLMs to TSC but does not change LLM
> > > - **COSLight KDD’24:** Grouped observation clustering and shared-policy MARL for TSC, but no change to MARL
> > > - **GPLight+ TEC’25:** leverages genetic programming for TSC, but no change to GP
> > > - **Proactive-XLight TMC’25:** Use predicted traffic as feature for MARL, but no change to MARL
> > > - **GPLight IJCAI’23:** clustered signals with shared-policy MARL for TSC, but no change to MARL
> > > - **Advanced-MP/CoLight ICML’22**: finds pressure and demand are the most useful feature as observation
> > > - **CoLight CIKM’19:** applies GAT and RL to multi-intersection TSC, but no change to GAT or RL
> > >
> > > Secondly, regarding policy-aware, it is also novel in TSC. Because the traffic condition is also influenced by the trained signal policy, only relying on the observed traffic condition is not sufficient and could lead to the mismatch between embeddings and policies. Therefore, we capture it explicitly by introducing reward into observation. As evidenced by the above list, the existing solutions only focus on the observation scope and feature, but did not takes the policy (reward) as part of observation, so they cannot be policy-aware and their performance suffers.
> > >
> > > Finally, regarding the intersection mapping, if a pure location-specific lookup mechanism, then an intersection with a fixed ID would be mapped to the same group regardless of how its traffic changes. This is exactly the problem we are solving: by **identity together with the current history query**, the same intersection now can be routed differently when its recent traffic evolution changes. Therefore, giving identity to the assignment module does not by itself mean that the learned groups are merely location-specific lookup decisions rather than shared behavioral structure.
> > >
> > > Given these clarifications and strong empirical evidence, we sincerely ask the reviewer to reconsider the novelty and impact of our work and please raise your score accordingly.

---

### Official Review · Reviewer_mihN · 2026-03-13

**Soundness:** 3
**Presentation:** 3
**Significance:** 3
**Originality:** 3
**Overall Recommendation:** 4
**Confidence:** 3

**Summary:**

This work proposes SLight, a scalable shared-policy framework for multi-agent traffic signal control that decouples reusable policy learning from non-stationary traffic patterns. By encoding each intersection with a policy-aware traffic pattern embedding and adaptively assigning it to one of K learned control principles, SLight enables efficient sharing under heterogeneity and improves network-level efficiency at scale. The authors conduct extensive experiments on real and synthetic networks to evaluate the effectiveness, scalability, and interpretability of SLight, revealing improvements over existing methods.

**Compliance With Llm Reviewing Policy:**

Affirmed.

**Final Justification:**

The author's response effectively addressed my concerns.  I am inclined to accept this work.

**Key Questions For Authors:**

Please answer the questions mentioned in the **Weaknesses**, such as scalability and more comparison methods.

**Limitations:**

Yes

**Strengths And Weaknesses:**

**Strengths:**
1. This work is well-organized and easy to understand.
2. The proposed method is technically feasible and performs well on several datasets.
3. The authors have open-sourced the code anonymously.


**Weaknesses:**
1. The authors state that the proposed method has good scalability, but it was only validated on  datasets with a maximum of 196 intersections. The data scale is insufficient, making the experiments less convincing.

2. As shown in Figure 5a, on the largest dataset NY, only four policy models are needed to achieve the best results. Are each policy model equally adopted by the intersection agent? Are there any policy models that are likely to be used very infrequently? On small datasets, is a few policy models, or even just one, sufficient?

3. In Table 2, the compared methods are all outdated; all of them were published three years ago. The author should compare the proposed method with more advanced methods.

---

> ### Author Rebuttal · Authors · 2026-03-31
>
> We sincerely thank the reviewer for the detailed and constructive comments. We are encouraged that you found the paper **clear and well-written**, with **sufficient evaluation metrics** and **helpful hyperparameter analysis for reproducibility**. We hope our response addresses your concerns, and we are happy to clarify any remaining questions.
> ### **[W1] Scalability**
>
> Thank you for your insightful comment.
> We agree that the original evaluation up to 196 intersections was insufficient for a central scalability claim. We have now added results on the Manhattan network with 2,510 intersections, which extends the evaluation to a substantially larger metropolitan-scale setting.
> As shown in the added table, SLight achieves best average travel time (ATT) on this larger network against representative rule-based and RL-based baselines. Due to rebuttal-time constraints, we currently report ATT as the end-to-end metric on Manhattan.
>
> | **Model** | **Manhattan** |
> | --- | --- |
> | Fixed-Time | 1844.56 |
> | MaxPressure | 1441.23 |
> | Advanced-Colight | 1138.66 |
> | Advanced-Mplight | 1374.73 |
> | CoSlight | 1053.90 |
> | **Slight** | **856.94** |
>
> ### **[W2] Policy Distribution**
>
> Thanks for your insightful observation! Yes, in NY, the learned policies are all meaningfully used rather than collapsing to a single dominant policy. We have added a **principle distribution** analysis to show the usage of each of the four shared policies and the **traffic patterns** associated with them , as illustrated in **Figure 1** of the anonymous repository: `https://anonymous.4open.science/r/SlightRebuttal-CC66/`.
>
> | Principle ID | Decision count | Global usage ratio |
> | --- | --- | --- |
> | 0 | 1043 | 26.61% |
> | 1 | 901 | 22.98% |
> | 2 | 982 | 25.05% |
> | 3 | 994 | 25.36% |
> - Policy 0 is used most often overall. From the prototype plot, it corresponds to the heaviest traffic state among the four policies: both queue and pressure are highest, especially in the north-south direction, and the north-south queue is clearly larger than the east-west queue.
> - Policy 1 is used least often overall. Its prototype shows relatively light traffic, with the smallest east-west queue and east-west pressure, while the north-south queue is still noticeably larger than the east-west queue.
> - Policy 2 corresponds to a medium-to-heavy traffic state. Both queue and pressure are above Policy 1 and Policy 3, and it still shows a north-south bias, but the directional gap is smaller than in Policy 0 and Policy 1.
> - Policy 3 corresponds to the most balanced traffic state. Its total load is not the highest, and the gap between north-south and east-west traffic is the smallest, so it is closer to a balanced control mode than the other three policies.
>
> For smaller datasets, one shared policy is not sufficient. Performance improves when using a small number of shared policies rather than only one. We verify this by varying the number of shared policies from 1 to 4 on Jinan-1, as shown in the added table.
>
> | Dataset/principle number | 1 | 2 | 3 | 4 |
> | --- | --- | --- | --- | --- |
> | JN-1 | 264.72 | 260.43 | 248.32 | 246.14 |
> ### **[W3] Newer Baselines**
>
> Thank you for carefully reviewing our paper and raising this point.
> We have now added recent RL-based baselines, including CoSLight[KDD’24], GPLight+[TEC’25], Proactive-XLight[TMC’25], and LLM-based TSC baselines. As shown in the added table, Slight achieves promising performance under the comparison of Rule-based, Unstructured RL-based, Grouped RL-based and LLM-based methods.
> We report average travel time (ATT) as the evaluation metric because it directly reflects end-to-end traffic efficiency. For LLM-based methods, we report full-shot results on Jinan and Hangzhou, which are the settings that were reproducible within the rebuttal period. We do not report NY or larger-network results for these methods because public model weights and training pipelines are unavailable, and reproducing them is prohibitively expensive within the rebuttal timeline.
>
> | Model | JN1 | JN2 | JN3 | HZ1 | HZ2 | NY1 |
> | --- | --- | --- | --- | --- | --- | --- |
> | CoSlight | 391.63 | 269.93 | 289.40 | 389.30 | 441.66 | 1404.99 |
> | Transformerlight | 256.16 | 235.78 | 232.39 | 273.18 | 313.91 | 1097.33 |
> | Proactive-Colight | 431.07 | 435.14 | 416.86 | 433.16 | 470.64 | 1024.62 |
> | Gplight+ | 329.14 | 278.94 | 287.10 | 312.99 | 410.19 | 1181.42 |
> | Llama3.3-70B | 272.41 | 244.55 | 243.53 | 281.44 | 326.42 | - |
> | Qwen 2.5-72B | 275.42 | 251.41 | 264.21 | 282.13 | 329.34 | - |
> | GPT 3.5-turbo | 337.32 | 328.19 | 343.19 | 293.42 | 348.59 | - |
> | GPT-4o | 281.58 | 259.61 | 258.85 | 280.48 | 325.48 | - |
> | DeepSeek-R1-671B | 279.11 | 258.43 | 262.21 | 278.56 | 335.53 | - |
> | DeepSeek-R1-Distill-7B | 331.45 | 311.43 | 288.42 | 291.32 | 344.73 | - |
> | Traffic-R1-3B | 270.34 | 239.53 | 238.03 | 277.83 | 324.11 | - |
> | **Slight** | **246.14** | **234.01** | **230.90** | **272.12** | **304.11** | **870.41** |

---

> > ### Author Rebuttal · Reviewer_mihN · 2026-04-05
> >
> > The author's response effectively addressed my concerns. It would be best to add these experimental analyses to the final version.

---

> > > ### Author Response · Authors · 2026-04-07
> > >
> > > Thank you very much for your thoughtful reconsideration and for your positive feedback. We sincerely appreciate that our response has effectively addressed your concerns.
> > >
> > > We fully agree that these additional experimental analyses should be included in the final version, and we will make sure to incorporate them clearly in the revised manuscript to further strengthen the paper.

---

### Official Review · Reviewer_mky5 · 2026-03-13

**Soundness:** 3
**Presentation:** 3
**Significance:** 3
**Originality:** 3
**Overall Recommendation:** 4
**Confidence:** 4

**Summary:**

This paper focuses on the scalable traffic signal control problem, and proposes a policy-aware grouped MARL-TSC framework, SLight. It extracts policy-influenced traffic patterns, learns group-level shared control principles and matches traffic pattern embedding. Multiple experiments shows that SLight achieves SOTA performance and demonstrates good scalability.

**Compliance With Llm Reviewing Policy:**

Affirmed.

**Final Justification:**

The rebuttal has well addressed my concerns on limited baselines, related works and dataset size. I will rise my score to 4.

**Key Questions For Authors:**

**[Q1]** Due to the limited baselines, could you provide additional new baselines for traffic signal control? The results on LLM-based TSC also need to be provided to prove the claim - LLM-based TSC cannot achieve SOTA without zero-shot settings.

**[Q2]** Due to the limited related works, please include more recent and relevant papers in the related work section.

**[Q3]** Due the limited dataset size, could you include larger-scale datasets in the experiments? The GPU footprint or training time also need to be provided to prove the efficiency of the method.

**Limitations:**

Authors didn't discuss the limitations and potential negative societal impact of the work.

The potential limitations of this work include the limited consideration of heterogeneous intersection structures. The potential negative societal impact of the work may include the safety concern of the traffic network.

**Strengths And Weaknesses:**

**Strengths:**

**[S1]** This paper is well-written and easy to understand with sufficient definition clarifications.

**[S2]** This paper provides sufficient evaluation metrics to judge the effectiveness of the method.

**[S3]** Multiple experimental results on hyperparameters are introduced in the experiment part, providing key parameter settings for reproducibility.

**Weaknesses:**

**[W1] Limited baselines**

Based on the experimental results provided by the authors, the conclusion that SLight shows better performance than baselines cannot be supported. Since the experimental comparison appears incomplete. The baselines considered in the experiments are limited to methods published before 2023, and there are several new baselines for traffic signal control tasks, such as CoSLight [1], GPLight+ [2], Proactive-XLight [3], etc., which are not included in the experiment.

Besides, the related work section mentioned that LLM-based TSC cannot achieve SOTA without zero-shot settings. The results on LLM-based TSC need to be provided to prove this claim.

**[W2] Limited related works**

Although this paper provides detailed introduction about traffic signal control task classification, the related works mentioned on MARL TSC are limited and outdated. Recent works somehow related to this work, including MetaVIM [4], TransformerLight [5], CoSLight [1], VLMLight [6], etc., should be included.

**[W3] Limited dataset size**

This work claims that SLight can be adopted in large-scale datasets and achieve scalability and efficiency balance. However, the number of intersections in datasets is only up to 196, while the datasets (Manhattan) used in MPLight [] contains 2510 intersections. Larger-scale datasets should be included in the experiments.

Besides, the GPU footprint or training time should be provided to prove the efficiency of the method.

**Reference:**

[1] Ruan, Jingqing, et al. "Coslight: Co-optimizing collaborator selection and decision-making to enhance traffic signal control." *Proceedings of the 30th ACM SIGKDD Conference on Knowledge Discovery and Data Mining*. 2024.

[2] Liao, Xiao-Cheng, Yi Mei, and Mengjie Zhang. "GPLight+: a genetic programming method for learning symmetric traffic signal control policy." *IEEE Transactions on Evolutionary Computation* (2025).

[3] Jiang, Yang, et al. "Proactive-XLight: Proactive traffic signal control with pluggable and reliable traffic prediction." *IEEE Transactions on Mobile Computing* (2025).

[4] Zhu, Liwen, et al. "Metavim: Meta variationally intrinsic motivated reinforcement learning for decentralized traffic signal control." *IEEE Transactions on Knowledge and Data Engineering* 35.11 (2023): 11570-11584.

[5] Wu, Qiang, et al. "Transformerlight: A novel sequence modeling based traffic signaling mechanism via gated transformer." *Proceedings of the 29th ACM SIGKDD conference on knowledge discovery and data mining*. 2023.

[6] Wang, Maonan, et al. VLMLight: Safety-critical traffic signal control via vision-language meta-control and dual-branch reasoning architecture. *Proceedings of the Thirty-Ninth Annual Conference on Neural Information Processing Systems* (2025).

---

> ### Author Rebuttal · Authors · 2026-03-31
>
> We sincerely thank the reviewer for the detailed and constructive comments. We are encouraged that you found the paper **clear and well-written**, with **sufficient evaluation metrics** and **helpful hyperparameter analysis for reproducibility**. We hope our response addresses your concerns, and we are happy to clarify any remaining questions.
>
> ### **[W1&Q1] Limited Baselines**
>
> Thank you for identifying these new baselines. To validate the better performance of SLight, we have tested more recent baselines (CoSLight[KDD’24], GPLight+[TEC’25], Proactive-XLight[TMC’25], and LLM-based TSC baselines). As shown in the table below, Slight still achieves the best performance and scalability. Specifically, we evaluate over Average Travel Time (ATT) because it directly reflects end-to-end traffic efficiency. For LLM-based methods, we report full-shot results on Jinan and Hangzhou, which are the settings that were reproducible within the rebuttal period. We do not report NY or larger networks for them because public model weights and training pipelines are unavailable, and reproducing them is prohibitively expensive within rebuttal period.
>
> | Model | JN1 | JN2 | JN3 | HZ1 | HZ2 | NY1 |
> | --- | --- | --- | --- | --- | --- | --- |
> | CoSlight | 391.63 | 269.93 | 289.40 | 389.30 | 441.66 | 1404.99 |
> | Transformerlight | 256.16 | 235.78 | 232.39 | 273.18 | 313.91 | 1097.33 |
> | Proactive-Colight | 431.07 | 435.14 | 416.86 | 433.16 | 470.64 | 1024.62 |
> | Gplight+ | 329.14 | 278.94 | 287.10 | 312.99 | 410.19 | 1181.42 |
> | Llama3.3-70B | 272.41 | 244.55 | 243.53 | 281.44 | 326.42 | - |
> | Qwen 2.5-72B | 275.42 | 251.41 | 264.21 | 282.13 | 329.34 | - |
> | GPT 3.5-turbo | 337.32 | 328.19 | 343.19 | 293.42 | 348.59 | - |
> | GPT-4o | 281.58 | 259.61 | 258.85 | 280.48 | 325.48 | - |
> | DeepSeek-R1-671B | 279.11 | 258.43 | 262.21 | 278.56 | 335.53 | - |
> | DeepSeek-R1-Distill-7B | 331.45 | 311.43 | 288.42 | 291.32 | 344.73 | - |
> | Traffic-R1-3B | 270.34 | 239.53 | 238.03 | 277.83 | 324.11 | - |
> | **Slight** | **246.14** | **234.01** | **230.90** | **272.12** | **304.11** | **870.41** |
> ### **[W2&Q2] Limited Related Works**
>
> Thank you for identifying these related works to make the background discussion more complete. We have now expanded it to include recent MARL-based and LLM/VLM-based TSC methods, including MetaVIM, TransformerLight, CoSLight, GPLight+, Proactive-XLight, and VLMLight.
>
> Specifically, in the 2) RL-based TSC discussion, we will add: “**MetaVIM [4] introduced a meta-RL framework for decentralized traffic signal control under changing environments; TransformerLight [5] explored gated Transformer-based sequence modeling for traffic signal control; CoSLight [1] co-optimized collaborator selection and decision-making to improve coordination efficiency; GPLight+ [2] used genetic programming to learn symmetric traffic signal control policies; and Proactive-XLight [3] incorporated reliable traffic prediction for proactive traffic signal control**.”
>
> In the  3) **LM**-based TSC discussion, we will add: “**VLMLight [6] further introduced vision-language meta-control and dual-branch reasoning for safety-critical traffic signal control**.”
> ### **[W3&Q3] Limited Dataset Size**
>
> Thanks for pointing this out. To validate the scalability, we have added new experiments on on  Manhattan network with 2,510 intersections. As shown in the table below, SLight achieves best ATT on this larger network against representative rule-based and RL-based baselines. Due to rebuttal-time constraints, we currently report ATT as the end-to-end metric on Manhattan.
>
> | **Model** | **Manhattan 2.5K** |
> | --- | --- |
> | Fixed-Time | 1844.56 |
> | MaxPressure | 1441.23 |
> | Advanced-Colight | 1138.66 |
> | Advanced-Mplight | 1374.73 |
> | CoSlight | 1053.90 |
> | **Slight** | **856.94** |
>
> We have also provided training and inference time of 50th epoch, as well as GPU footprint, to make the efficiency discussion more complete.
>
> | Model | Dataset | Training time (s) | Peak GPU memory (MB) | Inference time(s) | Peak GPU memory (MB) |
> | --- | --- | --- | --- | --- | --- |
> | Slight | New York | 2604.79 | 18950 | 0.022668 | 41.64 |
>
> ### **Limitation Discussion**
>
> Thanks for providing these limitation discussions. The limitation of this work is the generalization to transfer the learned policies across cities, as we aim to optimize the policies for each network specifically. The potential negative societal impact may result in deteriorating the navigation app’s routing accuracy and user experience, which relies on the stable traffic signal period to let the driver know the red light time, estimate travel time, and plan the routes. Nevertheless, this is the impact of all the dynamic TSC on navigation.

---

> > ### Author Rebuttal · Reviewer_mky5 · 2026-04-03
> >
> > The rebuttal has well addressed my concerns on limited baselines, related works and dataset size. I will rise my score to 4.

---

> > > ### Author Response · Authors · 2026-04-07
> > >
> > > Thank you very much for your careful reconsideration and for increasing the score to **4**. We sincerely appreciate your thoughtful evaluation and are very glad that our rebuttal has adequately addressed your concerns regarding the baselines, related work, and dataset scale.

---

### Decision · Program_Chairs · 2026-04-30

**Decision:**

Accept (regular)

**Comment:**

This paper proposes SLight, to learn group-level shared control principles by matching the obtained policy-influenced traffic patterns. Multiple experiments shows that SLight achieves SOTA performance and demonstrates good scalability, together with released code.

The authors successfully persuaded most reviewers to raise their scores. I would like to follow them to make the accept recommendation.